# New Conjugates Based on AIS/ZnS Quantum Dots and Aluminum Phthalocyanine Photosensitizer: Synthesis, Properties and Some Perspectives

**DOI:** 10.3390/nano12213874

**Published:** 2022-11-02

**Authors:** Dmitry Yakovlev, Ekaterina Kolesova, Svetlana Sizova, Kirill Annas, Marina Tretyak, Victor Loschenov, Anna Orlova, Vladimir Oleinikov

**Affiliations:** 1Shemyakin-Ovchinnikov Institute of Bioorganic Chemistry, Russian Academy of Science, 117997 Moscow, Russia; 2Prokhorov General Physics Institute, Russian Academy of Science, 119991 Moscow, Russia; 3Faculty of Photonics, ITMO University, 197101 Saint Petersburg, Russia

**Keywords:** quantum dots, aluminum phthalocyanine, photosensitizer, FRET, photodynamic therapy, reactive oxygen species (ROS)

## Abstract

Today, fluorescent diagnostics and photodynamic therapy are promising methods for diagnosing and treating oncological diseases. The development of new photosensitizers (PS) is one of the most important tasks to improve the efficiency of both laser-induced diagnostics and therapy. In our study, we conjugated PS with AIS/ZnS triple quantum dots (QDs) to obtain non-aggregated complexes. It was shown that the conjugation of PS with QDs does not change the PS fluorescence lifetime, which is a marker of the preservation of PS photophysical properties. In particular, efficient resonant Förster energy transfer (FRET), from QDs to PS molecules in the conjugate, increases the PS luminescence response. The FRET from QD to PS molecules with different ratios of donor and acceptors are shown. It has been demonstrated that the average efficiency of FRET depends on the ratio of PS and QD and reaches a maximum value of 80% at a ratio of 6 PS molecules per 1 QD molecule. Thus, these studies could help to contribute to the development of new complexes based on QD and PS to improve the efficiency of phototheranostics.

## 1. Introduction

Semiconductor zero-dimensional nanocrystals, called quantum dots (QDs), have attracted considerable interest in recent years, especially in connection with their application for fluorescent diagnostics of various types of oncological diseases [1,2]. This is a new class of fluorescent biological labels with unique optical properties compared to conventional organic dyes, widely used in clinical practice. Conventional organic photosensitizers have low photostability, narrow absorption spectra, and broad emission spectra, which limits their effectiveness in long-term imaging and multiplexing [3]. The use of QDs in the biological environment for fluorescent tumor diagnostics has shown that the unique properties of QDs can overcome these shortcomings [4]. The distinctive properties of QDs for biomedical spectroscopy and imaging include high photoluminescence quantum yield, very high molar extinction coefficients (10–100 times greater than organic dyes), broad absorption, covering the range from ultraviolet to near infrared, and large effective Stokes shifts, which make it easier to separate the excited and emitted light. An additional advantage of QDs is their resistance to metabolic degradation, and they are more photostable than traditional fluorescent agents used for fluorescent diagnostics [5]. In comparison with photosensitizers in the molecular form, QDs have such a special property that their fluorescence spectra can be controlled by size and composition [6]. Unfortunately, the often-studied quantum dots of II-VI or III-V groups of the periodic table, containing “class A” metals, such as Cd, As, Hg and Pb, are not used in biological research due to their toxicity [7,8,9,10,11,12,13,14]. In this regard, it seems promising to develop and study biomarkers for tumor diagnostics based on QDs from relatively non-toxic semiconductors of groups I-III-VI, such as indium and silver sulfide AgInS2 (AIS). QDs themselves are capable of generating only reactive oxygen species or a small amount of singlet oxygen (^1^O_2_), which, in comparison with classical photosensitizers (PSs), are inferior in efficiency in laser-induced therapy. However, QDs are able to non-radiatively transfer energy to photosensitizer (PS) molecules, thereby increasing the generation of ^1^O_2_ PS, as well as increasing the concentration of the organic dye in the tumor tissue due to high extinction coefficient of QDs in a broad spectral range [15], which, when conjugated, can increase the efficiency of photodynamic therapy (PDT) [16]. In this case, it should be taken into account that the luminescence spectrum of QDs should overlap as best as possible with the absorption spectrum of PS for possibility of Förster resonant energy transfer (FRET) to PS molecules.

To date, various classes of photosensitizers are used for fluorescence diagnostic (FD) and PDT of oncological and inflammatory diseases, one of which is aluminum phthalocyanine-arginine (AlPc), with the commercial name Photosens. This PS has appeared positively in the diagnosis and treatment of many cancers [17,18]. According to the literature data, the quantum yield of the generation of singlet oxygen of AlPc is quite high, and amounts to 0.42 ± 0.06 [19,20,21]. In addition, colloidal QDs also have the advantage of a low elimination time from the body [22]. It was demonstrated in [23] that, on average, the half-life of QDs from the body of mice is no more than one week after a single intravenous injection. These characteristics may also increase the effectiveness of therapy due to the absence of the need for constant PS injections before the cure for multiple FD and PDT.

Group I–III–VI triple QDs, such as QDs AIS and AIS/ZnS, are considered good candidates for bioimaging because they do not contain toxic elements, have a high absorption coefficient [24], a long luminescence lifetime [25,26,27,28], and quantum yield of the fluorescence is more than 15% [29,30]. Moreover, the synthesis of new complexes based on AIS/ZnS will eliminate the necessity for an individual selection of the source of exciting radiation. Complexes based on AIS/ZnS triple quantum dots and PS (AlPc) have not been studied yet. The evaluation of the spectral and fluorescent properties of conjugates based on AIS/ZnS QDs and PS is important for their use in fluorescent diagnostics of tumor tissue, as well as to study the conjugation method, determine the optical properties of AIS/ZnS QDs and PS, and evaluation of the efficiency of non-radiative energy transfer from QD to PS with different ratios.

## 2. Materials and Methods

### 2.1. QDs Synthesis

We used Ag-In-S2 and Ag-In-S2/ZnS colloidal quantum dots synthesized according to [31]. First, Ag-In-S2 cores were synthesized, then a ZnS shell was grown to them. QD samples were taken from the common solution at different synthesis times, which are listed in Table 1, thus varying the thickness of the ZnS shell on the AIS cores.

### 2.2. Used Equipment

The absorption and luminescence spectra of AIS/ZnS QDs colloidal solutions were obtained using a UV-3600 spectrophotometer (Shimadzu, Kyoto, Japan) and a Cary Eclipse spectrofluorimeter (Varian, Palo Alto, CA, USA). The hydrodynamic radius of AIS/ZnS QDs was obtained by dynamic light scattering (DLS) using a Zetasizer Nano ZS analyzer (Malvern instruments Ltd., Worcestershire, UK). Sizes of AIS/ZnS QDs were obtained using a Zeiss Merlin scanning electron microscope (SEM) (Carl Zeiss AG, Oberkochen, Germany). The analysis of the kinetics of the studied samples was carried out using a fluorescent microscope, equipped with a detector operating in the single photon counting mode MicroTime100 (Pico Quant, Berlin, Germany). The absorption and luminescence spectra of colloidal QDs and conjugates were analyzed using a UV probe 3600 spectrophotometer (Shimadzu, Kyoto, Japan) and a Cary Eclipse spectrofluorimeter (Varian, Palo Alto, CA, USA). Magnetic circular dichroism spectra were recorded with a Jasco J-1500 CD spectrophotometer using an MCD-581 magnetic attachment at a field strength of −1.5, 0, 1.5 T.

Distilled water was used as a solvent. The molar concentration of QDs in three conjugates was 33 µM. The molar concentration of AlPc in conjugate 1 and 3 was 20.4 µM, and in conjugate 2, it was 10.2 µM. For these conjugates the molar ratios of QDs and PS were determined by spectroscopic method.

Aluminum phthalocyanine-arginine (AlPc) was kindly provided by the Scientific Research Institute of Organic Intermediates and Dyes of JSC “NIOPIK”.

## 3. Results and Discussion

### 3.1. Conjugation of AIS/ZnS QDs and PS Molecules

The synthesis of QDs conjugates with aluminum phthalocyanine molecules was carried out according to the scheme described below. To 65 µL of AIS/ZnS (5×10^−5^ M) was added 65 µL of 2-(N-morpholino)ethanesulfonic acid (MES) (pH 5) and 35 µL of N-hydroxysulfosuccinimide (sulfo-NHS) in MES (21.7 mg/mL), incubated for 30 min with constant stirring and at room temperature. Then 3.5 μL of 3-(ethyliminomethyleneamino)-N,N-dimethylpropan-1-amine (EDC) in MES (19.2 mg/mL) was added and incubated for another 60 min with constant stirring and room temperature. Activated AIS/ZnS was purified from unreacted components of the mixture using Amicon Ultra 30 kDa centrifugal concentrators for 7 min at 10,000 rpm. After that, the volume of activated AIS/ZnS was adjusted to 100 µL and 200 µL of bicarbonate buffer (BCB), pH 9.6, was added. A solution of AlPc was prepared, the structural formula of which is shown in Figure 1: a sample was dissolved in 10 µL of 0.1 M NaOH, 90 µL of H_2_O was added. The concentration was determined by a spectroscopic method. The calculated amount of AlPc was added to the activated AIS/ZnS with the molar ratio as 3:1, 6:1, respectively. Conjugates were incubated for 2 h with constant stirring and at room temperature. The resulting conjugates were purified using Amicon Ultra 30 kDa centrifugal concentrators for 7 min at 10,000 rpm, the supernatant liquid was collected, 200 µL of BCB, pH 9.6 was added (the procedure was repeated 2 times). To the purified conjugate (50 µL) was added 50 µL of BCB. The resulting conjugates were stored at +4.0 °C.

### 3.2. Characterization of Ag-In-S_2_/ZnS QDs

Table 1 lists the hydrodynamic diameter of QDs obtained by DLS. For QDs with and without shell, size analysis was performed using a scanning electron microscope.

With the help of SEM, the dimensions of QDs during a 34-min build-up of the shell were obtained with dimensions of 17.2 ± 0.7, which coincides with the data obtained by the DLS method.

The spectroscopic properties of AIS/ZnS QDs were studied. Figure 2 shows the absorption and fluorescence spectra of the samples.

Figure 2a shows that the absorption of QDs is observed in the range of 300–500 nm, while there is no distinct absorption peak. A decrease in the absorption capacity of QDs can be caused due to the presence of a large number of defect states within the band gap, which cause the Urbach tail in the absorption spectrum of QDs, which is attributed to phonon-assisted optical transitions [32,33]. As can be seen from Figure 2b, the growth of the ZnS shell is accompanied by a short-wavelength shift of the QDs luminescence spectra, with a maximum at 580 nm, and a significant increase in its intensity. In this case, the width of the luminescence band is 100 nm. This process is related to the fact that the shell eliminates defects on the surface of the QDs core, mainly “dangling bonds” arising from undercoordinated metal atoms forming chalcogenide vacancies. At the same time, the shell enhances the retention of photogenerated charge carriers in the QDs core, inhibiting their interfacial transfer and promoting radiative recombination. A slight shift of the luminescence maximum towards higher energies is due to the inclusion of Zn cations in the lattice of the AIS nucleus, which leads to an increase in the band gap in the optical range. Both effects are characteristic of the studied AIS/ZnS QDs [34].

The QD fluorescence decay kinetics were estimated as a function of the ZnS shell growth time. Figure 3 shows the fluorescence decay curves of AIS and AIS/ZnS 20 nm QDs, respectively. Approximation of parameters are shown in Table 2.

According to the results obtained from Table 2, shell growth leads to an increase in the characteristic QDs fluorescence decay times, which correlates with the fluorescence spectra (Figure 2) and literature data [32].

### 3.3. Spectral and Luminescent Characteristics of Conjugates Based on AIS/ZnS QDs and PS

The aggregation of PS molecules in conjugates leads to loss of their ability to luminesce. In addition, Figure 4 includes characteristic spectral manifestations of molecular aggregation in conjugate 1 [35]. Thus, the decrease in the amplitude of the Soret band at a wavelength of 350 nm, as well as the electronic Q-band at a wavelength of 675 nm and its vibrational band at 610 nm. In the absorption spectrum of PS in the conjugate with QDs, a clearly distinguishable absorption band appears in the spectral region (about 650 nm), which is attributed in the literature to PS H-dimers [36,37]. Weak absorption bands in the spectral regions shorter than 590 nm and longer than 720 nm toward low energies, as well as broadening of the red edge of the Q bands, indicate the presence of PS J dimers and/or higher order aggregates in complexes with QDs [37]. It can be assumed that the aggregation of PS in complexes with QDs occurs along the same pathway as the formation of aggregates in aqueous solutions of PS due to the formation of a chemical bond between the central aluminum atoms of PS “oxygen bridge” (Al-O-Al bonds), since this is one of the most probable ways of aggregation of these molecules [38].

To assess the efficiency of the formation of complexes between QDs and PS, a comparative analysis of the absorption spectra of these conjugates was carried out. The PS absorption spectra in the conjugates are shown in Figure 5.

The luminescent properties of the obtained conjugates based on AIS/ZnS QDs, and PS AlPc were studied. For conjugate 1, conjugate 2, and conjugate 3, the molar ratios of QDs and PS correspond as: 1:6, 1:3, 1:6, respectively, which were determined by spectroscopic method. Figure 5 shows the absorption spectra of the conjugates, as well as the absorption spectra of free QDs and PSs.

To confirm the absence of AlPc aggregation in the conjugates, based on AIS/ZnS QDs and PS molecules, the magnetic circular dichroism (MCD) spectra were analyzed for all samples (Figure 6).

The MCD spectra of QDs comprise two opposite-signed symmetrical bands, centered at 667 nm and 682 nm due to Zeeman splitting of the Q-band, due to degenerate state caused by the high symmetry of the molecule [39]. Data on magnetic circular dichroism indicate the absence of aggregates in sample solutions, since a comparative analysis of the spectral curves of pure PS with conjugates does not show any changes in the shapes of the spectra themselves according to identification of bands in the MCD spectrum in work [40].

An analysis was made of the fluorescence spectra of the samples at different excitation wavelengths, shown in Figure 6. The fluorescence spectrum upon excitation at 600 nm allows to observe the fluorescence of PS molecules in conjugates (Figure 7b). Excitation at a wavelength of 485 nm (Figure 8) makes it possible to assess the presence of FRET from QDs to PS molecules, and, in this case, AlPc luminescence is observed; however, at this wavelength, free AlPc practically does not absorb in this region compared to QDs. 

The luminescence spectra of conjugates with PS molar concentrations of 20.4 μM for conjugates 1 and 3, 10.2 μM for conjugate 2, and free PS with a molar concentration of 29.5 μM) upon excitation at 600 nm, have been compared. Here, the luminescence intensity is directly proportional to the PS concentrations in the conjugates, which indicates that the luminescence quantum yield of the PS in the conjugates is preserved. It was found that the luminescence quantum yield of the PS in the conjugates does not change compared to the free PS. The luminescence spectrum of the free PS at a wavelength of 485 nm was subtracted from the PS luminescence spectra in the conjugates, which is related to the intrinsic absorption of the PS at this wavelength, and the dependence of the sensitized PS luminescence on the PS concentrations in the conjugates was obtained. According to the data obtained, shown in Figure 8, it can be concluded that in conjugates numbered 1, 2, 3, a resonant energy transfer from QDs to PS is clearly observed, since upon excitation of the samples (λex = 485 nm), QDs absorb radiation, and the luminescence spectra exhibit a distinct peak at 675 nm, which is characteristic of PS. However, if the resonant transfer did not occur, then we would register the luminescence shoulder of QDs, according to Figure 2b. Upon excitation (λex = 600 nm), the PS luminescence spectrum is clearly observed. The luminescence lifetime of the obtained conjugates was measured in comparison with the fluorescence lifetime of pure PS, which dissolved in 0.1% NaOH aqueous solution (Figure 8).

An analysis was carried out of the fluorescence decay kinetics of free PS and PS conjugated with QDs, synthesized at different times. In this case, to record the quenching time of the PS luminescence in conjugation with the QDs, an interference long pass filter was used, which singles out the maximum of the PS luminescence. The obtained results of the luminescence decay of the studied samples were approximated by an exponential function (Formula (1)) and the approximation parameters, characteristic decay times of the samples and their amplitudes are presented in Table 3.
(1)$$f\left(t\right)=A*\;e^{-t/\tau }+C$$

This analysis showed that the interaction of QDs with PS does not affect the fluorescence lifetime of free PS, while the amplitude ratio shows the influence of the AIS QDs shell thickness.

### 3.4. Calculation of Theoretical and Experimental FRET

An analysis was made of the effectiveness of theoretical FRET with QDs on PS molecules (Figure 9) according to the formulas:(2)$$Q_{FRET}=\frac{R_{0}^{6}}{R_{0}^{6}+R^{6}},$$
(3)$$R_{0}^{6}=\frac{9000*ln10*\;\Phi^{2}*q_{0D}}{128*\pi^{5}*n^{4}*N}\int I_{D}^{H}\left(v\right)*\;\varepsilon_{A}\left(v\right)*\;v^{-4}*dv,$$
where $q_{0D}$ is the luminescence quantum yield of the donor in the absence of an acceptor; $\Phi^{2}$—orientation factor; $I_{D}^{H}*\left(v\right)$—normalized quantum spectral density of donor luminescence ($\int I_{D}^{H}\left(v\right)*dv=1$), $\varepsilon_{A}\left(v\right)$—coefficient of molar acceptor extinction spectrum; $\left(v\right)$—wave number; *n*—refractive index of the medium; and N—Avogadro’s number.

The theoretical analysis using Formulas (2) and (3) showed that in the case of one QD, with an average diameter of 16 nm per one PS molecule, the efficiency of non-radiative transfer is 93%; with an increase in the number of PS molecules to 6, the theoretical FRET increases to 97%. In conjugation of 1 QD with an average diameter of 20 nm and one PS molecule, the efficiency of non-radiative energy transfer is 82%, when one QD is conjugated with three PS molecules, the theoretical FRET increases to 92%, and when one QD is conjugated with six PS molecules, the theoretical FRET increases to 96% (Table 4).

The calculation of the experimental efficiency of energy transfer from QDs to PS molecules was carried out according to Formula (4):(4)$$E=\frac{D_{A}\left(\lambda_{ex}\right)*I_{AD}}{D_{D}\left(\lambda_{ex}\right)*I_{A}}$$
where $I_{AD}$ and $I_{A}$ are the acceptor luminescence intensities in the presence and absence of a donor, respectively, upon excitation at the same wavelength; $D_{D}$ and $D_{A}$ are the optical densities of the donor and acceptor at the donor luminescence excitation wavelength.

In this regard, for conjugates 1 and 3 in a ratio of 1:6, the experimental energy transfer is observed ~ 80% (Q_theor._ = 96%), against a conjugate 2 in a ratio of 1:3, where FRET is 48% (Q_theor._ = 92%). The loss of transfer efficiency is due to the fact that additional channels are created that compete with FRET. 

In this regard, it can be said that with an increase in the number of acceptors on the donor surface, the efficiency of non-radiative energy transfer also increases. 

## 4. Conclusions

For the first time, a method for the complexation of conjugates based on semiconductor QDs AIS/ZnS and AlPc was proposed, and their optical properties were studied. Optical absorption spectra are used to show and describe the characteristic differences between aggregated PS molecules in conjugates, which lose their ability to luminesce, and conjugated molecules. It was shown that the method of analysis of conjugates using MCD also confirms the absence of aggregates. It was found that the fluorescence lifetime of free PS does not differ from the fluorescence lifetime of PS in complex with QDs; the luminescent properties of the PS monomer conjugated with the QD are retained, as well as the photophysical properties of the PS, in particular, the luminescence quantum yield.

The developed method of conjugation makes it possible to bind up to six PS molecules with QDs, retaining their monomeric form, and also makes it possible to significantly increase the amount of singlet oxygen compared to one QD: one PS complexes and the changing the size of the QDs does not affect to the efficiency of energy transfer.

The theoretical and experimental efficiency of FRET from QDs to PS molecules with different numbers of acceptors is shown. Thus, for conjugates with a molar ratio of QD and PS of 1:6, the highest efficiency of non-radiative energy transfer is on average 80% (the theoretical efficiency for conjugate 1 and conjugate 2 is on average 96%), and in the conjugate with a ratio of 1:3, the efficiency of energy transfer reduced by 32% (theoretical efficiency—82%). It is assumed that these studies may open up new approaches to the creation of QD-based complexes for the generation of singlet oxygen with a number of potential applications.

## Figures and Tables

**Figure 1 nanomaterials-12-03874-f001:**
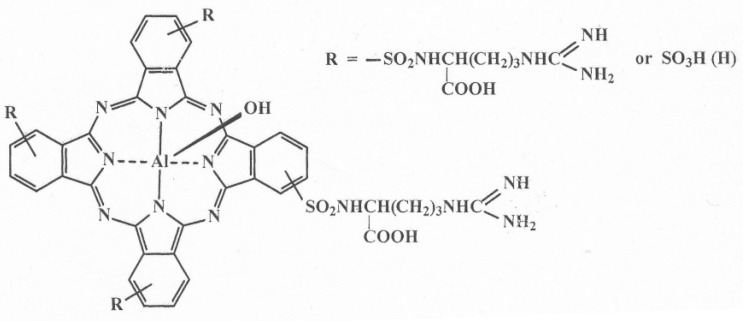
Structural formula of PS.

**Figure 2 nanomaterials-12-03874-f002:**
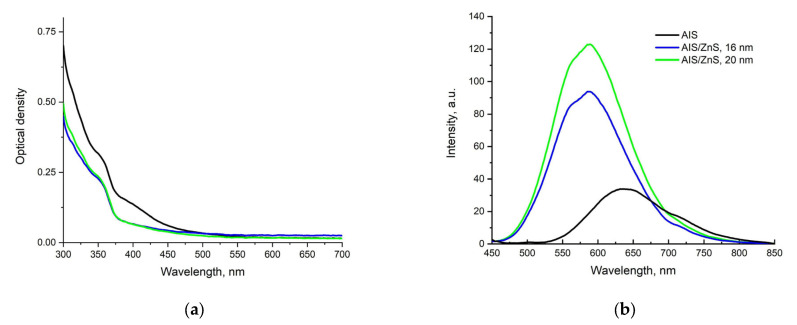
(**a**) The absorption spectra of QDs in an aqueous solution, (**b**) The fluorescence spectra of QDs in an aqueous solution, excitation with light with a wavelength of 430 nm.

**Figure 3 nanomaterials-12-03874-f003:**
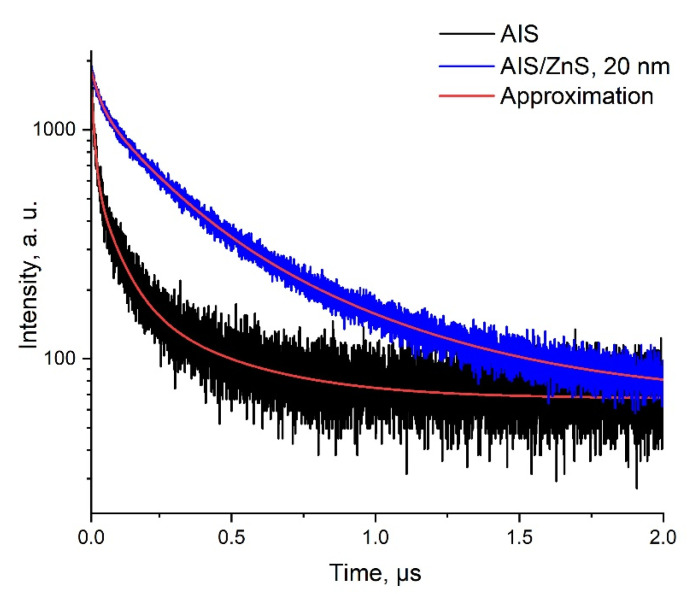
Fluorescence decay curves of AIS cores QDs and fluorescence decay curves of AIS/ZnS QDs 20 nm. Triexponential function approximation: $f\left(x\right)=A_{1}*e^{-\tau_{1}/t}+A_{2}*e^{-\tau_{2}/t}+A_{3}*e^{-\tau_{3}/t}$.

**Figure 4 nanomaterials-12-03874-f004:**
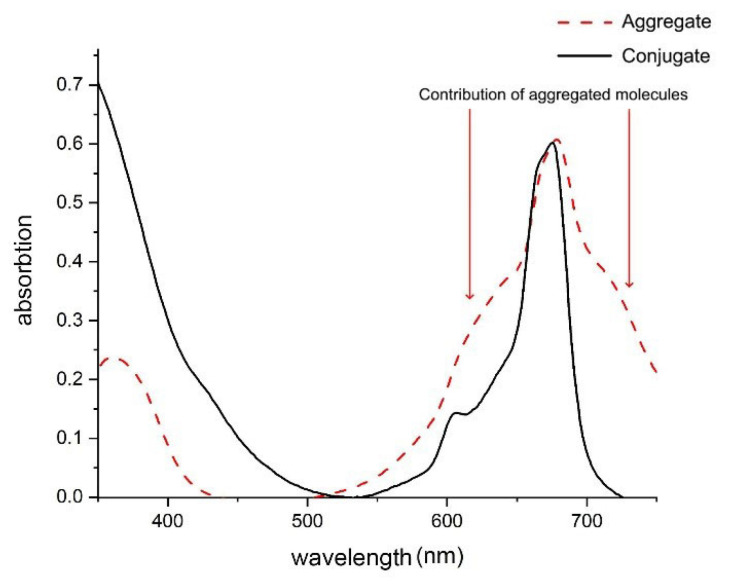
Comparison of the absorption spectra of the conjugate 1 and the aggregate. The samples were dissolved in distilled water. The molar concentration of QDs in the conjugate was 33 µM. The molar concentration of AlPc in conjugate 1 was 20.4 µM.

**Figure 5 nanomaterials-12-03874-f005:**
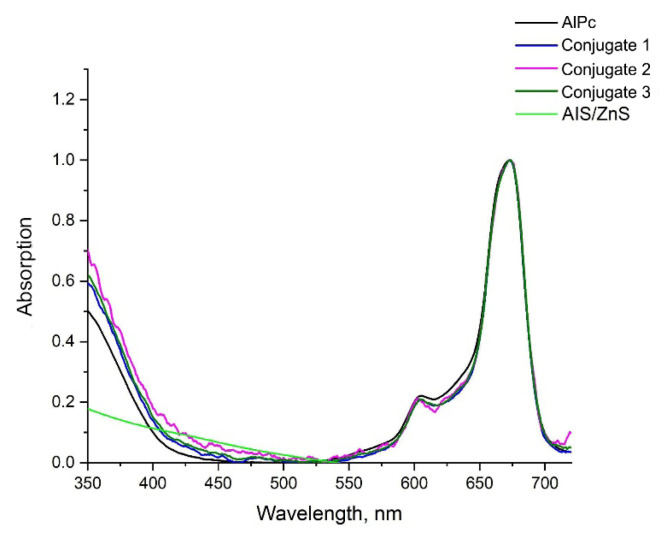
Absorption spectra of the studied samples, normalized to the maximum (λ = 660 nm). The samples were dissolved in distilled water. The molar concentration of QDs in three conjugates was 33 µM. The molar concentration of AlPc in conjugate 1 and 3 was 20.4 µM, in conjugate 2 was 10.2 µM.

**Figure 6 nanomaterials-12-03874-f006:**
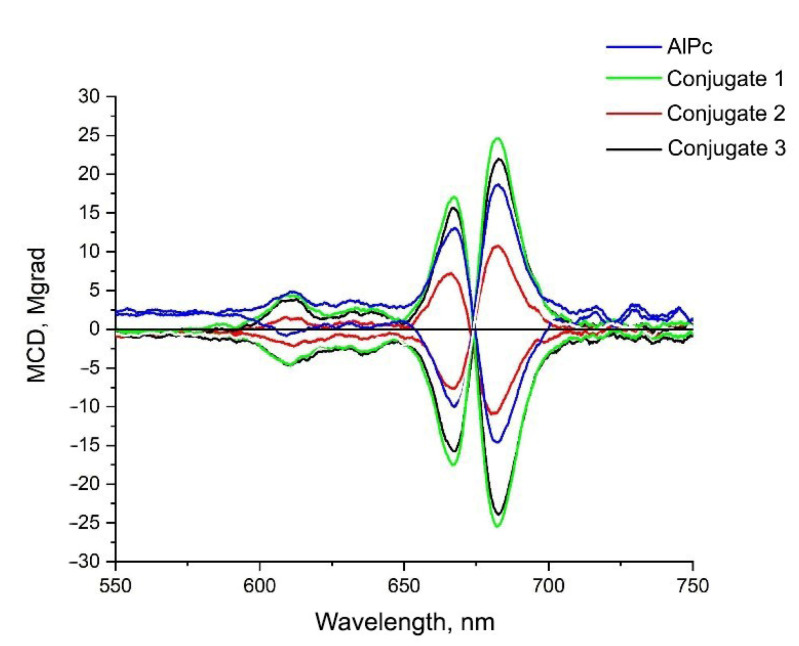
Spectra of magnetic circular dichroism of the studied samples. The samples were dissolved in distilled water. The molar concentration of QDs in three conjugates was 33 µM. The molar concentration of AlPc in conjugate 1 and 3 was 20.4 µM, in conjugate 2 was 10.2 µM. MCD-581 magnetic attachment was at a field strength of −1.5, 0, 1.5 T.

**Figure 7 nanomaterials-12-03874-f007:**
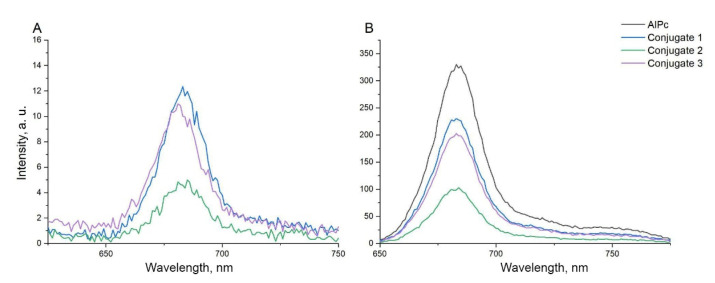
(**a**) Fluorescence spectrum of samples (λex = 485 nm), (**b**) Fluorescence spectrum of samples (λex = 600 nm). The molar concentration of QDs in three conjugates was 33 µM. The molar concentration of AlPc in conjugate 1 and 3 was 20.4 µM, in conjugate 2 was 10.2 µM.

**Figure 8 nanomaterials-12-03874-f008:**
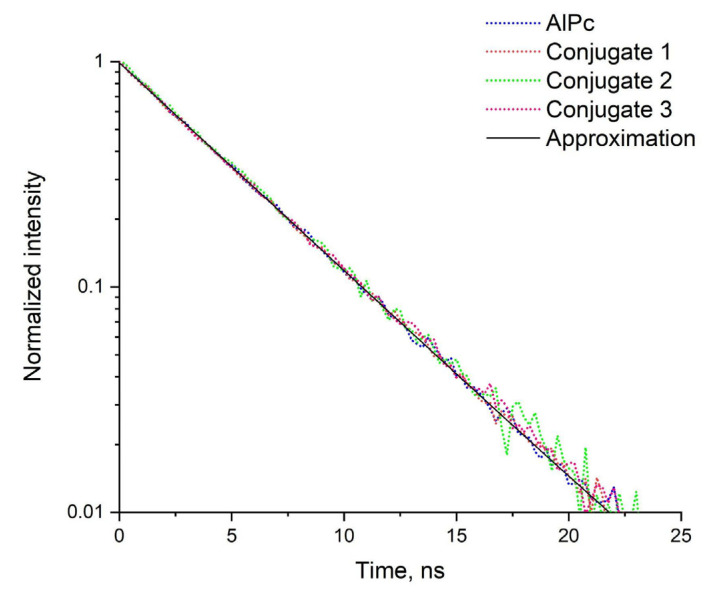
Luminescence decay curves of the conjugates and free AlPc. The molar concentration of QDs in three conjugates was 33 µM. The molar concentration of AlPc in conjugate 1 and 3 was 20.4 µM, in conjugate 2 was 10.2 µM. The signal was recorded in the spectral range from 600 to 800 nm.

**Figure 9 nanomaterials-12-03874-f009:**
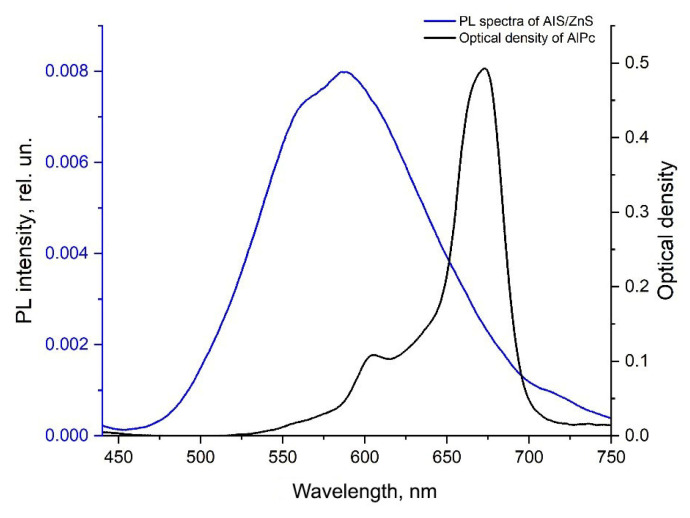
Overlapping spectra of AIS/ZnS QDs luminescence and PS absorption spectrum marked by green. The donor luminescence spectrum (QDs) was normalized by area.

**Table 1 nanomaterials-12-03874-t001:** Results of QDs size estimation using the Zetasizer Nano ZS analyzer.

Sample	Shell Synthesis Time, min.	Average Hydrodynamic Diameter, nm
AIS	0	6
AIS/ZnS	24	16
AIS/ZnS	34	20

**Table 2 nanomaterials-12-03874-t002:** Results of QDs AIS and AIS/ZnS the fluorescence decay kinetics.

Sample	τ_1_, ns.	Contribution of the First Component, %	τ_2_, ns.	Contribution of the Second Component, %	τ_3_, ns.	Contribution of the Third Component, %
AIS	348 ± 35	58	80 ± 8	34	11 ± 1	8
AIS/ZnS 20 nm	566 ± 55	61	203 ± 20	34	29 ± 3	5

**Table 3 nanomaterials-12-03874-t003:** Values of the characteristic luminescence decay times of the samples under study.

Sample	τ, ns	I, kCounts
AlPc	4.73	192
Conjugate 1	4.73	195
Conjugate 2	4.73	80
Conjugate 3	4.78	145

**Table 4 nanomaterials-12-03874-t004:** Values of theoretical and experimental FRET depending on the ratio of donor and acceptors in the samples.

Sample	QD Size, nm	QD: AlPc Ratio	Q_theor._, %	Q_exp._, %
Conjugate 1	16	1:6	97	81
Conjugate 2	20	1:3	92	48
Conjugate 3	16	1:6	96	80

## Data Availability

The authors confirm that the data supporting the findings of this study are available within the article.

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
