# Peer review of "New Conjugates Based on AIS/ZnS Quantum Dots and Aluminum Phthalocyanine Photosensitizer: Synthesis, Properties and Some Perspectives"

_nanomaterials, 2022, doi:10.3390/nano12213874_

Round 1

Reviewer 1 Report

Line 89: "5*10-5M" should be changed to "5*10-5M"

Author Response

We greatly appreciate the critical reading of our manuscript and suggested modifications.

Comment: Line 89: "5*10-5 M" should be changed to "5*10-5 M"

Respond: Thanks for the note, the mistake has been corrected in the text of the Manuscript.

Reviewer 2 Report

The authors addressed to develop a conjugated PS with 15 AIS/ZnS triple QDs to obtain non-aggregated complexes to overcome the shortcomings of PS. The aim to improve the efficiency of phototheranostics. This might be interesting for readers and to additional investigations. I suggest few corrections into the text.

Line 53: what’s FSs? Please you clarify the acronym.

Line 101: Please clarify the ratio FA : AIS/ZnS. To what refers 1:1 and to 6:1? I suggest using round brackets to distinguish them.

Line 102: maybe you meant “at room temperature”.

Line 125 and 127: it lacks the company of the mentioned instruments.

Line 130: I suggest reformulating the sentence.

The Results and discussion capitol number 3 is completely missing. The authors should insert it and also the enumeration (e.g. 3.1…3.5).

Line 179: maybe the authors have confused the ratio (1:6; 1:1; and again 1:6?)

Author Response

We greatly appreciate the critical reading of our manuscript and suggested modifications. We thoroughly revised the paper introducing your remarks along with these of other Reviewers.

Comment: Line 53: what’s FSs? Please you clarify the acronym.

Respond: The typo has been corrected in the text of the Manuscript

Comment: Line 101: Please clarify the ratio FA : AIS/ZnS. To what refers 1:1 and to 6:1? I suggest using round brackets to distinguish them.

Respond: Sorry for the confusion. The sentence has been modified.

Comment: Line 102: maybe you meant “at room temperature”.

Respond: Thank you for the note, the mistake has been corrected.

Comment: Line 125 and 127: it lacks the company of the mentioned instruments.

Respond: The companies have been mentioned for all equipment in the Manuscript.

Comment: Line 130: I suggest reformulating the sentence.

Respond: The sentence has been reformulated.

Comment: The Results and discussion capitol number 3 is completely missing. The authors should insert it and also the enumeration (e.g. 3.1…3.5).

Respond: The enumeration of sections has been added to the Manuscript.

Comment: Line 179: maybe the authors have confused the ratio (1:6; 1:1; and again 1:6?)

Respond: Thank you for the note, the difference between conjugate 1 (molar ratios of QDs and AlPc correspond as: 1:6) and conjugate 3 (molar ratios of QDs and AlPc correspond as: 1:6) is that was used QDs with different ZnS shell formation times.

Reviewer 3 Report

The manuscript by Yakovlev et al. reports on the synthesis of AIS/ZnS quantum dots that were conjugated to aluminum phthalocyanine photosensitizer to obtain non-aggregated complexes that might be used in therapeutic applications. The overall work lacks to the originality. In addition many aspects in this manuscript were missed or only briefly discussed. Therefore, I cannot accept the article in this present form and I recommend that article should be rewritten to show really the interest of such system. In the following comments, I will explain the weak points of the manuscript and the reasons of my decision in the chronological order of appearance in the manuscript.

1.      I have a problem with the abbreviation in the whole manuscript: while sometimes they are wrongly attributed, in another sections of the manuscript the abbreviations are not defined.

2.      The title does not correspond to the content, since the authors did not show any phototheranostic applications?

3.      Please define the abbreviations at the first time of appearance in the text. For example (FD, PDT, PS…) in the abstract. Please check the whole manuscript in this perspective.

4.      In the abstract: please remove “does not reduce the therapeutic effect…” since no data on this subject were shown in the manuscript.

5.      In the introduction, authors are talking about quantum dots (QDs), so the “CTs” abbreviation corresponds to what?

6.      “Classical FSs” should be classical PSs.

7.      Authors claimed in the following sentence that “However, AIS are able to nonradiatively transfer energy to photosensitizer (PS) molecules….. aluminum phthalocyanine for more efficient Festor resonant energy transfer (FRET) to PS molecules.” In fact, I have a problem with this sentence, how the nonradiative emission of the QDs will increase the generation of singlet oxygen of the PS, all the more so authors are saying in the following sentence that the fluorescence of the QDs should match with the absorption of the PS. Thus in this case the radiative emission of the QDs enables the excitation of the PS and hence the generation of the PSs? This sentence should be corrected/ reformulated.  

8.      “Förster” not “Festor Resonnance Transmission”

9.      What authors want to say by this sentence: “At the same time, it has a number of advantages, one of  which is the slow removal from the study area.” Which removal the authors are talking about? Is it from the human body? If so then it should be the “elimination” process. Please correct this sentence. “Numerous” not “number”.

10.  Please avoid redundancy in the introduction section, as for example “The aim of this work ... The purpose of this work is to study the conjugation method…”.

11.  5×10-5 M: Please write “-5” as superscript.

12.  Could author precise how, the Photosense-arginine (FA) was synthesized? A scheme about the synthetic route will be very helpful. And how the solution of this PS was prepared?

13.  Page 4: Authors claimed that “Thus, the decrease in the amplitude of the Soret band at a wavelength of 350 nm”, however the spectra in the figure 4 start from 400 nm. Thus authors should show the corresponding spectra. In addition, authors compared the absorption spectrum of the conjugates to that of aggregates. Could authors precise how these aggregates were prepared?

14.  Page 5: Authors used three conjugates that were referred to “For conjugate 1, conjugate 2, conjugate 3, the molar ratios of QDs and PS correspond as: 1:6, 1:1, 1:6”. I think there is a problem here. Please check the ratio.

15.  In figure 5, the Y axis refers to the absorbance so it should be dimensionless (a.u) and not (cm-1). Also One should use absorbance instead of optical density when showing an absorption spectrum.

16.  Page 7: “that in conjugates numbered 2, 3, 4”: should be 1,2, 3?

17.  In figure 8: authors should specify in which solvent the PS was solubilized? Is it in water so in the form of aggregates or it was soluble? Please show the corresponding absorption spectrum in the SI.

18.  Authors mentioned in the conclusion but also in the abstract that “the quantum yield of singlet oxygen generation, the luminescent properties of the PS monomer conjugated with QDs are retained, and the photochemical properties of the PS, in particular, the quantum yield of singlet oxygen generation, are demonstrated with a high probability”. I do not understand this statement, all the more so the authors did not determine the singlet oxygen yield in the whole manuscript? So how the authors came to such a conclusion?

Author Response

We highly appreciate the valuable remarks of the Reviewer and proposed suggestions. We tried to comply with all raised by the Reviewer questions and as a matter of fact we considerably modified the paper.

Comment: I have a problem with the abbreviation in the whole manuscript: while sometimes they are wrongly attributed, in another sections of the manuscript the abbreviations are not defined.

Respond: Thank you very much for this remark. The abbreviations have been corrected and changed to the same style.

Comment: The title does not correspond to the content, since the authors did not show any phototheranostic applications?

Respond: The title has been corrected and corresponds to the purpose of the Manuscript.

Comment: Please define the abbreviations at the first time of appearance in the text. For example (FD, PDT, PS…) in the abstract. Please check the whole manuscript in this perspective.

Respond: Thank you very much for your navigation. My colleagues advised me to add abbreviations in the Abstract, and also to add abbreviations in the text of the Manuscript, because usually the abstract is read separately from the whole article.

Comment: In the abstract: please remove “does not reduce the therapeutic effect…” since no data on this subject were shown in the manuscript.

Respond: The sentence has been modified.

Comment: In the introduction, authors are talking about quantum dots (QDs), so the “CTs” abbreviation corresponds to what?

Respond: Thank you for this comment. The abbreviation has been corrected in the Manuscript.

Comment: “Classical FSs” should be classical PSs.

Respond: The mistake has been corrected.

Comment: Authors claimed in the following sentence that “However, AIS are able to nonradiatively transfer energy to photosensitizer (PS) molecules….. aluminum phthalocyanine for more efficient Festor resonant energy transfer (FRET) to PS molecules.” In fact, I have a problem with this sentence, how the nonradiative emission of the QDs will increase the generation of singlet oxygen of the PS, all the more so authors are saying in the following sentence that the fluorescence of the QDs should match with the absorption of the PS. Thus in this case the radiative emission of the QDs enables the excitation of the PS and hence the generation of the PSs? This sentence should be corrected/ reformulated.

Respond: In the case of QDs/PS molecule complexes, an efficiency of generation of singlet oxygen can be significantly enhanced as compared with molecular photosensitizers due to high extinction coefficient of QDs in a broad spectral range. According to the information, the sentence has been reformulated.

[Maslov, V., Orlova, A., & Baranov, A. (2011). Combination Therapy: Complexing of QDs with tetrapyrrols and other dyes. In Photosensitizers in Medicine, Environment, and Security (pp. 351-389). Springer, Dordrecht.]

Comment: “Förster” not “Festor Resonnance Transmission”

Respond:The mistake has been corrected.

Comment: What authors want to say by this sentence: “At the same time, it has a number of advantages, one of which is the slow removal from the study area.” Which removal the authors are talking about? Is it from the human body? If so then it should be the “elimination” process. Please correct this sentence. “Numerous” not “number”.

Respond: Thank you for the note, the typo has been corrected in the text of the Manuscript.

Comment: Please avoid redundancy in the introduction section, as for example “The aim of this work ... The purpose of this work is to study the conjugation method…”.

Respond: We modified the Introduction section.

Comment: 5×10-5 M: Please write “-5” as superscript.

Respond: Thanks for the note, the mistake has been corrected in the text of the Manuscript

Comment: Could author precise how, the Photosense-arginine (FA) was synthesized? A scheme about the synthetic route will be very helpful. And how the solution of this PS was prepared?

Respond:  Aluminum phthalocyanine-arginine was kindly provided by the Scientific Research Institute of Organic Intermediates and Dyes of (Russia). PS was dissolved in 0.1% NaOH aqueous solution.

Comment: Page 4: Authors claimed that “Thus, the decrease in the amplitude of the Soret band at a wavelength of 350 nm”, however the spectra in the figure 4 start from 400 nm. Thus authors should show the corresponding spectra. In addition, authors compared the absorption spectrum of the conjugates to that of aggregates. Could authors precise how these aggregates were prepared?

Respond: Thank you for the note, another absorption  spectrum has been added. The formation of PS aggregates in a complex with quantum dots mainly occurs due to the formation of a chemical bond between the central Al atoms of the “oxygen bridge” (Al–O–Al bonds), since for this type of PS this is one of the two most probable ways of aggregation in aqueous solutions. At the same time, the aggregated PS molecules lose their ability to luminesce.

Comment: Page 5: Authors used three conjugates that were referred to “For conjugate 1, conjugate 2, conjugate 3, the molar ratios of QDs and PS correspond as: 1:6, 1:1, 1:6”. I think there is a problem here. Please check the ratio.

Respond: The difference between conjugates 1 (molar ratios of QDs and PS correspond as: 1:6) and conjugates (molar ratios of QDs and PS correspond as: 1:6) 3 is that they use QDs with different ZnS shell formation times.

Comment: In figure 5, the Y axis refers to the absorbance so it should be dimensionless (a.u) and not (cm-1). Also One should use absorbance instead of optical density when showing an absorption spectrum.

Respond: The mistake has been corrected.

Comment: Page 7: “that in conjugates numbered 2, 3, 4”: should be 1,2, 3?

Respond: The typo has been corrected in the text of the Manuscript.

Comment: In figure 8: authors should specify in which solvent the PS was solubilized? Is it in water so in the form of aggregates or it was soluble? Please show the corresponding absorption spectrum in the SI.

Respond: Thank you for the note, the typo has been corrected in the text of the Manuscript. PS was presented as a molecular solution, the absorption spectrum of which is shown in Figure 4.

Comment: Authors mentioned in the conclusion but also in the abstract that “the quantum yield of singlet oxygen generation, the luminescent properties of the PS monomer conjugated with QDs are retained, and the photochemical properties of the PS, in particular, the quantum yield of singlet oxygen generation, are demonstrated with a high probability”. I do not understand this statement, all the more so the authors did not determine the singlet oxygen yield in the whole manuscript? So how the authors came to such a conclusion?

Respond: Thank you for your careful reading of the manuscript. In the Results section we have described in more detail the data on which it was concluded that the quantum yield and luminescent properties were preserved. In conclusion, we have corrected the proposal on the generation of singlet oxygen in order not to mislead the readers.

Round 2

Reviewer 3 Report

The authors have addressed all the questions in the revised version of the manuscript. Thus, I recommend the publication of the manuscript in its current form.

Author Response

We highly appreciate and thank you for your careful reading of our manuscript and the proposed modifications.